# Evaluating Forest Site Quality Using the Biomass Potential Productivity Approach

Xingrong Yan [1,2], Linyan Feng [2], Ram P. Sharma [3], Guangshuang Duan [2,4], Lifeng Pang [2], Liyong Fu [2] and Jinping Guo [1,*]

1    College of Forestry, Shanxi Agricultural University, Jinzhong 030800, China; b20201057@stu.sxau.edu.cn
2    Research Institute of Forest Resource Information Techniques, Chinese Academy of Forestry, Beijing 100091, China; linyan_feng@caf.ac.cn (L.F.); duans@xynu.edu.cn (G.D.); plf619@ifrit.ac.cn (L.P.); fuly@ifrit.ac.cn (L.F.)
3    Institute of Forestry, Tribhuwan University, Kathmandu 46000, Nepal; ramsharm1@gmail.com
4    College of Mathematics and Statistics, Xinyang Normal University, Xinyang 464000, China
*    Correspondence: guojinping@sxau.edu.cn; Tel.: +86-139-3540-6926

**Abstract:** Biomass productivity is of great significance for the evaluation of forest quality, which is important for the improvement of forest management. We propose the computational methods of biomass potential productivity (BPP) and biomass realistic productivity (BRP), both of which provide reliable practical guides for predicting forest growth under multi-aged, multi-species, and multi-layered canopy conditions. We used 2222 national forest inventory plots that were measured in four consecutive periods in the Jilin Province for this purpose. We analyzed and verified the computational methods of BPP based on the BRP and evaluated its practical significance. The results showed that growth models of the stand height, stand basal area, and stand biomass of four forest types (pure larch forest, larch broadleaf mixed forest, Mongolian oak pure forest, and Mongolian oak broadleaf mixed forest) fit adequately, BPP was greater than BRP, and this difference decreased with an increasing stand age, suggesting that the potential productivity of the middle-aged and young forest was higher than that of the mature forest, although the difference is minimal. In addition, the realistic productivity of stands with better site quality was close to the potential productivity, which is consistent with the biological significance of the potential productivity of the biomass. The degree of difference between the potential productivity of the biomass and the realistic productivity of biomass also decreases with the decline in site quality, and it can be termed as the potentially improved stand biomass. The BPP model was able to perform well in both the pure and mixed forests. The BRP not only verifies the rationality of the BPP but can be also used to quantify the forest site quality, which is helpful for evaluating forest growth and informed decision making in forestry.

**Keywords:** multi-aged mixed forest; site productivity evaluation; potentially improved stand biomass; site productivity modeling

## 1. Introduction

Site quality evaluation involves an assessment of the stand productivity potential of a forest stand with a certain method adopted, which is a necessary theoretical basis for the full utilization of the productive potential of the land and achieving scientific management of forest resources [1]. Since the 1920s, site type and site quality assessments have been receiving widespread attention [2]. So far, significant theoretical and technical progress has been made within research on forest site quality evaluation, and many evaluation methods have been proposed, such as the index plant method [3,4], the site index method [5,6], the growth index method [7–9], and the advanced exponential method [10]. On the one hand, traditional forest site quality evaluation is mainly aimed at plantation forests. For natural forests with a complex stand structure, an irregular matching of tree height class, age structure, and other factors, the method of forest site quality evaluation could be difficult

to meaningfully unity [11]. On the other hand, the current research methodology assesses only the realistic growth of the forest, which essentially reflects the real productivity of the stand. In contrast, it is difficult for forest sites to reach maximum productivity under natural conditions, as there could be disturbances caused by human and natural factors. Researchers have attempted to accurately evaluate forest quality by constructing ecological models, but data acquisition (climate, soil, etc.) problems increase the difficulty of subsequent large-scale applications [12,13]. The remote sensing approach, which helps identify and distinguish the stand features (species composition and structure, etc.) is also commonly applied for site quality assessment [14]. However, the results can be substantially affected by the environment and accuracy. The existing methods of site quality evaluation are still limited in practice, and thus, we need a comprehensive method that is suitable for evaluation of site quality on a large scale.

For assessing forest site quality on a large-scale, data availability is the basis for developing forest quality indicators. Since forest biomass is a major driver of forest structure and diversity, its accumulation or decrease determines the forest site quality, and hence, forest biomass can be used as a reliable indicator of site productivity [15–18]. Accurate computation of forest biomass productivity and its dynamics can have a reference value, which can be used for forest development, management, and utilization, and such an estimate may also solve the ecological problems [19–21]. The forest site quality depends on the forest and growth potential [22]. Biomass potential productivity (BPP) is the maximum annual biomass growth that can be achieved at a given stand age for the given site type and stand type [23].

To overcome the difficulties mentioned above, this study assumes that under the same site conditions and stand type (similar species composition) with a similar stand structure and stand density, there would be an approximately similar growth in the stand height, stand basal area, and stand biomass. Based on the concept of basal area potential productivity and volume potential productivity, this study proposes a biomass potential productivity that is applicable to multi-aged and multi-layered mixed forests [24,25]. The methods of forest site quality evaluation presented in this study can provide practical guidance for scientific forest management [26,27]. The ninth national forest resources inventory in the Jilin Province, which is one of the key provinces for forest development in China [28], shows that the forest area is 7.85 million hectares, and the standing volume is 101.296 million cubic meters. Thus, scientific evaluation of forest site quality in the Jilin Province is conducive to raising social awareness of improvements to the current state of forests. Forest site quality evaluation in the Jilin Province is considered very important, as this helps improve the prediction performance of future stand productivity and can also lay the methodological foundation for effective forest management strategies, the sustainable development of China's future forest ecosystems, and achieving carbon neutrality. This study aims to (1) propose a new forest site quality evaluation index: biomass potential productivity; (2) provide detailed computational methods of biomass potential productivity; (3) demonstrate the application of the computational methods using real data of the larch and Mongolian oak in the Jilin Province as an example; and (4) verify and validate the proposed methods and models. The methods and models proposed in this paper will provide both theoretical and practical knowledge that will be useful for forest site quality evaluation—one of the necessary informational inputs for scientific forest management.

## 2. Materials and Methods

### 2.1. Study Area

The forest area selected for this study lies in the northeast of China, which is a key forestry province (121°38′–131°19′ E, 40°50′–46°19′ N). The province has 45.2% forest cover, which includes 8,792,800 hectares of forest land and 1086 million cubic meters of accumulation. The reported mean annual temperature of the Jilin Province is 2–6 °C, and mean annual precipitation is 765.9 mm. The forest types are mainly medium-aged

temperate coniferous and broad mixed forests, with a wide variety of tree species and forest site productivities involved.

*2.2. Data Materials*

We used Chinese national forest inventory (NFI) data from the Jilin Province, collected in 1999, 2004, 2009, and 2014, which consist of 2222 plots that are distributed across the four main multi-aged and mixed-species stand types. Of the 2222 plots, 384 were allocated in pure larch forest (stand type I), 258 in larch broadleaf mixed forest (stand type II), 821 in Mongolian oak pure forest (stand type III), and 759 in Mongolian oak broadleaf mixed forest (stand type IV). The distribution of the sample plots is shown in Figure 1.

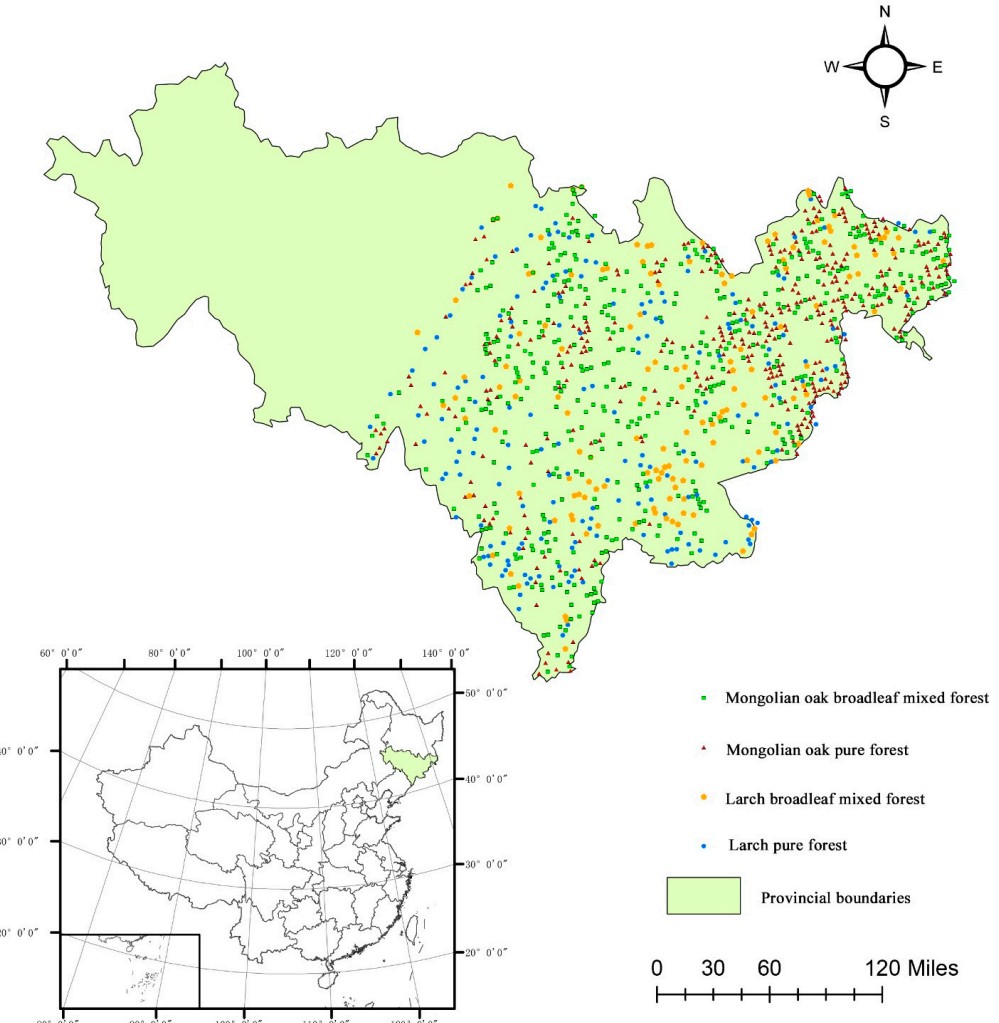

**Figure 1.** Overview of the study area and NFI plot locations.

*2.3. Methods*

2.3.1. Computing Sample Plot-Level Variables

We determined the number of trees, mean breast height, basal area, stand density index, biomass per plot, and stand density based on the basic NFI data. The lowest breast height diameter of a tree is >5 cm. The individual tree biomass was estimated using the method suggested by the State Forestry Administration (LY/T2258-2014) [29]. That is, individual tree biomass is the sum of trunk, bark, branches, leaves, and below-ground biomass, and the total biomass of the sample plot is the sum of all the individual tree biomasses (Table 1).

We summarized the relevant stand characteristics and various indicators (Table 2).

**Table 1.** Computation of stand-level variables.

| Variable | Formula |
|---|---|
| Mean of the diameter at breast height (cm) | $D = \sqrt{\frac{1}{n}\sum D_i{}^2}$ |
| Number of trees per hectare | $N = \frac{n}{p_a}$ |
| Stand basal area (m² ha⁻²) | $BA = \left(\sum \pi D_i{}^2 / 40,000\right) / p_a$ |
| Stand density index | $S = N(D/20)^{1.605}$ |
| Stand biomass (t ha⁻²) | $B = \frac{B_p}{p_a}$ |

Note: $n$ is the number of the trees per sample plot; $D_i$ is the diameter at breast height; $P_a$ is the plot area (0.067 hectares); and $B_p$ is the total biomass per sample plot.

**Table 2.** Description statistics of stand variables by stand type.

| Stand Type | Variable | Min. | Max. | Mean | SD |
|---|---|---|---|---|---|
| **I** | Stand height (m) | 4.10 | 23.00 | 12.19 | 3.99 |
| | Stand age (year) | 8.00 | 67.00 | 28.60 | 11.88 |
| | Stand density index | 61.85 | 1057.62 | 456.77 | 221.52 |
| | Stand basal area (m² ha⁻²) | 10.43 | 259.46 | 58.89 | 35.79 |
| | Stand biomass (t ha⁻²) | 5.12 | 126.08 | 29.21 | 17.64 |
| **II** | Stand height (m) | 4.00 | 23.00 | 12.72 | 4.216 |
| | Stand age (year) | 7.00 | 68.00 | 30.62 | 13.70 |
| | Stand density index | 5.97 | 1006.8 | 509.83 | 237.48 |
| | Stand basal area (m² ha⁻²) | 0.11 | 150.03 | 27.31 | 27.21 |
| | Stand biomass (t ha⁻²) | 0.04 | 219.31 | 49.04 | 45.36 |
| **III** | Stand height (m) | 1.50 | 22.00 | 13.05 | 3.74 |
| | Stand age (year) | 5.00 | 163.00 | 62.58 | 26.69 |
| | Stand density index | 137.05 | 1670.96 | 694.60 | 265.11 |
| | Stand basal area (m² ha⁻²) | 3.43 | 55.05 | 20.09 | 8.43 |
| | Stand biomass (t ha⁻²) | 20.37 | 485.65 | 136.05 | 67.17 |
| **IV** | Stand height (m) | 4.00 | 24.00 | 14.27 | 3.58 |
| | Stand age (year) | 8.00 | 169.00 | 63.12 | 27.20 |
| | Stand density index | 149.98 | 1614.57 | 682.70 | 261.69 |
| | Stand basal area (m² ha⁻²) | 3.65 | 53.48 | 20.02 | 8.79 |
| | Stand biomass (t ha⁻²) | 20.38 | 421.27 | 135.87 | 70.54 |

Note: stand type I is the pure larch forest; stand type II is the larch broadleaf mixed forest; stand type III is the Mongolian oak pure forest; stand type IV is the Mongolian oak broadleaf mixed forest.

2.3.2. Site Grouping Based on Stand Height Growth

Due to the existence of different stand heights of the same age, we classified each stand type into the following five site groups based on the age range of the stand without considering the effects of environmental factors and management practices [24,30] as follows:

(1) Determination of age group for each plot:

We divided the age of the sample plots in 5-year intervals.

$$A_c = \text{int}(age/5) + 1 \tag{1}$$

$$A_c = \begin{cases} k \, if \, \min age + (k-1)Dage \le age < \min age + kDage & k = 1, 2, \cdots m-1 \\ m \, if \, \min age + (k-1)Dage \le age \le \max age & k = m \end{cases} \tag{2}$$

where $A_c$ is the age grouping of the plot; $age$ is the stand age of the plot; $\text{int}(age/5)$ is an integrating function with parameter $age/5$, it is a vectorial mode of operation with an interval of 5, so the initial value is set to 5; $m$ is the number of groupings across the age span of the sample ($m = 5$); $\min age$ is the minimum age; $\max age$ is the maximum age;

*Dage* is the differences between age groups $(\max age - \min age)/m$. After computation of (Equations (1) and (2)), a sample plot can only belong to one age group.

(2) Determination of the initial height group of each sample plot:

For each plot, the stand height hierarchy at the beginning of the period was determined in each age group to which it belongs.

$$H_i^l = \begin{cases} kif\min H_i^A + (k-1)DH_i^A \leq H_i < \min H_i^A + kH_i^A & k = 1, 2, \cdots n-1 \\ H_i^l = nif\min H_i^A + (k-1)DH_i^A \leq H_i < \max H_i^A & k = n \end{cases} \tag{3}$$

where $H_i$ is the stand height; $H_i^l$ is the stand height classification of the plot, which was assumed to be $l$; $\min H_i^A$ is the minimum stand height for the age group to which the plot belongs; $\max_i^A$ is the maximum stand height for the age group to which the plot belongs; $n$ is the stand height group $(n = 5)$; $DH_i^A$ is the difference in stand height within the age group to which the sample site belongs $(\max H_i^A - \min H_i^A)/n$.

(3) Determination of height-age model form:

$$H = f_h(Age|F, L) \tag{4}$$

where $f_h()$ is the height-age model; $H$ is the stand height; $Age$ is the stand age; $F$ is the stand type (I, II, III, IV); $L$ is the site group (1, 2, 3, 4, 5).

### 2.3.3. Biomass Potential Productivity and Realistic Productivity

Stand productivity includes both the realistic productivity and potential productivity. The biomass potential productivity ($BPP$) refers to the biomass increment hat seeks a maximum objective function from the given stand density index ($S$) interval under the conditions of growth of stand basal area and stand biomass corresponding to a given stand type ($F$) and the corresponding parameter estimates, age (expressed by $Age_0$) and site group ($L$):

$$BPP = f(Age_0, S, \hat{\Phi}_G, \hat{\Phi}_B|F, L)S \in [S_{\min}, S_{\max}] \tag{5}$$

where $Age_0$ is the specific stand age, $\hat{\Phi}_G$ and $\hat{\Phi}_B$ parameters for stand basal area and stand biomass growth models, respectively. $S^{\min}$ and $S_{\max}$ is the set a feasible region of $S$; $F$ is the stand type (I, II, III, IV); $L$ is the site group (1, 2, 3, 4, 5). The specific calculation steps are as follows:

**Step 1, we need to have 5 known conditions:**

(1) Given the site group ($L$) of the stand type ($F$). Site group ($L$) was known from the Section 2.3.2.
(2) Given the specific stand age to be computed ($Age_0$)
(3) Given the density index S search interval $[S_{\min}, S_{\max}]$, the feasible region of $S$ is assumed to vary from 30 to 3000 [24].
(4) Form of basal area growth model corresponding to the specific site group ($L$) of the stand type ($F$) is

$$G = f_G(Age, S, \hat{\Phi}_G|F, L) \tag{6}$$

where $G$ is the basal area growth model of the stand type ($F$); $Age$ is the specific stand age; $S$ is the stand density index of the specific stand age; $\Phi_G$ is the model parameter; $F$ is the stand type (I, II, III, IV); $L$ is the site group (1, 2, 3, 4, 5).
(5) Form of biomass growth model corresponding to the specific site group ($L$) of the stand type ($F$) is

$$B = f_B(Age, S, \hat{\Phi}_B|F, L) \tag{7}$$

where $B$ is the biomass growth model of the stand type ($F$); $Age$ is the specific stand age; $S$ is the stand density index of the specific stand age; $\hat{\Phi}_B$ is the parameter vector to be estimated; $F$ is the stand type (I, II, III, IV); $L$ is the site group (1, 2, 3, 4, 5).

**Step 2, finding a reasonable stand density index (*S*) that maximizes biomass annual growth (*BI*).**

(1) Given a feasible region of $S(S_{\min}, S_{\max})$, a stand age ($Age_0$), a stand type ($F$), a site group ($L$), the error term ($\varepsilon$), and iteration t = 1. Computing of initial values of stand density index ($S$) at 4 points ($S_1^{Age_0}, S_2^{Age_0}, S_3^{Age_0}, S_4^{Age_0}$) using the golden section method [24].

(2) By computing $B_0$ at $Age_0$ and $B_1$ at $Age_1 = Age_0 + 1$, we can get biomass annual growth ($BI_1^{Age_0}, BI_2^{Age_0}, BI_3^{Age_0}, BI_4^{Age_0}$).

$$BI = B_1 - B_0 = f_B(Age_1, S_1, \hat{\Phi}_B) - f_B(Age_0, S_0, \hat{\Phi}_B) \tag{8}$$

where $f_B$ is the biomass growth model; $S_1$ is the assumed stand density index at $Age_1$; $S_0$ is the assumed stand density index at $Age_0$.

(3) Finding the $S$ that made the $BI$ maximized.

**Step 3, Computing biomass potential productivity (*BPP*).**

$$BPP = (BI_2^{Age_0} + BI_3^{Age_0})/2 \tag{9}$$

where $BI_2^{Age_0}$ and $BI_3^{Age_0}$ are the $S_2^{Age_0}$ and $S_3^{Age_0}$ corresponds to the maximizes biomass growth.

The detailed procedure of the above *BPP* algorithm is shown in the Supplementary Information S1.

The biomass realistic productivity (*BRP*) is the continuous annual growth of the stand biomass computed from the stand age ($Age_0$), stand density index ($S$), stand type ($F$) and site groups ($L$), combined with the basal area growth model, and the biomass growth model. The computational procedures of *BRP* are similar to *BPP*. It should be noted that, unlike the *BPP* calculation, the stand characteristics, $S$ from field surveys, should be provided for the *BRP* calculation. The stand density index ($S_0$) for biomass realistic productivity in $Age_0$ is the measured data from the sample plots. Theoretically, the realistic productivity is lower or closer to the potential productivity. The formula is as follows:

$$BRP = BI_{R1} - BI_{R0} = f_B(Age_1, S_1, \hat{\Phi}_B | F, L) - f_B(Age_0, S_0, \hat{\Phi}_B | F, L) \tag{10}$$

where *BRP* is the biomass realistic productivity; $BI_{R1}$ is the realistic biomass annual growth at $Age_1$; $BI_{R0}$ is the realistic biomass annual growth at $Age_0$; $S_1$ is the stand density index at $Age_1$; $S_0$ is the realistic stand density index at $Age_0$. *F* is the stand type (I, II, III, IV); *L* is the site group (1, 2, 3, 4, 5).

The detailed procedure of the above *BRP* algorithm is shown in the Supplementary Information S2.

2.3.4. Computing the Available Potential Improved Stand Biomass

The potential for improving the biomass productivity is obtained from the difference between the potential biomass productivity and the realistic biomass productivity.

$$BPI = BPP - BRP \tag{11}$$

where *BPI* is the available improved potential stand biomass; *BPP* and *BRP* are the biomass potential productivity and biomass realistic productivity corresponding to a certain stand age.

All computations in this study were implemented in R 4.3.0.

### 2.3.5. Modeling and Parameter Estimation

The Chapman-Richards function was selected as a basic model for fitting stand height data [25], and the model form is:

$$H_{(F,L)} = 1.3 + a_{i(F,L)}\left[\left(1 - e^{-b_{i(F,L)}Age}\right)\right]^{c_{i(F,L)}} + \varepsilon_{(F,L)} \tag{12}$$

where $H_{(F,L)}$ is stand height to the $F^{th}$ stand type for the $L^{th}$ site group; $Age$ is the stand age for the plots; $a_{i(F,L)}$, $b_{i(F,L)}$, $c_{i(F,L)}$ are the parameters corresponding to the model; $\varepsilon_{(F,L)}$ is the error term.

The stand basal area growth model adopts the functional form by Chapman-Ricards [25]:

$$G_{(F,L)} = a_{j(F,L)}\left[1 - \exp\left(-b_{j(F,L)}(S/1500)^{c_{j(F,L)}}Age\right)\right]^{d_{j(F,L)}} + \varepsilon_{(F,L)} \tag{13}$$

where $G_{(F,L)}$ is the stand basal area to the $F^{th}$ stand type for the $L^{th}$ site group (m²·ha⁻²); $S$ is the stand density index; $Age$ is the stand age; $a_{j_{(F,L)}}$, $b_{j_{(F,L)}}$, $c_{j_{(F,L)}}$ and $d_{j_{(F,L)}}$ are the parameters corresponding to the model; $\varepsilon_{(F,L)}$ is the error term.

The biomass growth model we used here is exactly the same model form as the basal area growth model:

$$B_{(F,L)} = a_{k(F,L)}\left[1 - \exp\left(-b_{k(F,L)}(S/1500)^{c_{k(F,L)}}Age\right)\right]^{d_{k(F,L)}} + \varepsilon_{(F,L)} \tag{14}$$

where $B_{(F,L)}$ is the stand biomass to the $F^{th}$ stand type for the $L^{th}$ site group (t²·ha⁻²); $S$ is the stand density index; $Age$ is the stand age; $a_{k_{(F,L)}}$, $b_{k_{(F,L)}}$, $c_{k_{(F,L)}}$ and $d_{k_{(F,L)}}$ are the parameters corresponding to the model; $\varepsilon_{(F,L)}$ is the error term.

### 2.3.6. Model Evaluation

We estimated models (12), (13) and (14) using the R package and evaluated their fitting performance using the coefficient of determination ($R^2$) and root mean square error (RMSE).

$$R^2 = 1 - \frac{\sum\limits_{i=1}^{k}\sum\limits_{j=1}^{n_i}(y_{ij} - \hat{y}_{ij})^2}{\sum\limits_{i=1}^{k}\sum\limits_{j=1}^{n_i}(y_{ij} - \overline{y}_{ij})^2} \tag{15}$$

$$RMSE = \sqrt{\sum\limits_{i=1}^{k}\sum\limits_{j=1}^{n_i}(y_{ij} - \hat{y}_{ij})^2/n} \tag{16}$$

where: $y_{ij}$ is the jth actual value of the ith sample plot; the jth estimated value of the ith sample plot; $\overline{y}_{ij}$ is the mean $H$ or $BA$ or $B$ of the observations; k is the number of classifications; n is the number of sample plots in the kth category. n is the number of all sample plots.

### 2.3.7. Reference Age

Reference age refers to an age after which the growth of the stand height tends to be stable, and the age at which the mean growth of the stand height or timber volume is the largest. The reference age of mixed forests is determined by dominant tree species or tree species more frequently occurring in a stand. According to the LY/T 2415-2015 "Technical Regulations for the Preparation of Status Index Tables", the reference age for both larch and Mongolian oak are 30 years [31].

## 3. Results

### 3.1. Parameter Estimates for Growth Models of Stand Height, Basal Area, and Biomass

Based on the NFI data of four forest types in the Jilin Province, the height–age model, basal area growth model, and biomass growth model of forest stands models (12), (13), and (14) were fitted, and parameter estimates were obtained (Table 3). All the parameter estimates were significant ($p < 0.05$), and each model described more than 70% variations in the response variables of the interest (Table 4). The fitting effect of each growth model was good, and the results were feasible.

**Table 3.** Parameter estimates of growth models by stand type.

| Stand Type | Site Group | Model (12) | | | Model (13) | | | | Model (14) | | | |
|---|---|---|---|---|---|---|---|---|---|---|---|---|
| | | $a_i$ | $b_i$ | $c_i$ | $a_j$ | $b_j$ | $c_j$ | $d_j$ | $a_k$ | $b_k$ | $c_k$ | $d_k$ |
| I | 1 | 23.19 | 0.036 | 0.93 | 43.18 | 0.03 | 4.71 | 0.21 | 528.03 | 0.00001 | 10.09 | 0.11 |
| | 2 | 19.81 | — | — | 42.11 | — | — | — | 505.12 | — | — | — |
| | 3 | 16.79 | — | — | 40.86 | — | — | — | 493.13 | — | — | — |
| | 4 | 13.53 | — | — | 38.89 | — | — | — | 454.20 | — | — | — |
| | 5 | 9.78 | — | — | 37.69 | — | — | — | 430.16 | — | — | — |
| II | 1 | 22.58 | 0.05 | 1.34 | 105.14 | 0.00001 | 5.18 | 0.11 | 685.03 | 0.00002 | 10.02 | 0.13 |
| | 2 | 19.04 | — | — | 104.09 | — | — | — | 673.58 | — | — | — |
| | 3 | 16.48 | — | — | 100.27 | — | — | — | 645.42 | — | — | — |
| | 4 | 14.09 | — | — | 98.47 | — | — | — | 576.05 | — | — | — |
| | 5 | 9.88 | — | — | 97.28 | — | — | — | 549.29 | — | — | — |
| III | 1 | 20.22 | 0.03 | 1.08 | 96.76 | 0.00002 | 9.33 | 0.11 | 539.57 | 0.00007 | 13.99 | 0.08 |
| | 2 | 17.41 | — | — | 94.01 | — | — | — | 516.80 | — | — | — |
| | 3 | 15.11 | — | — | 92.26 | — | — | — | 498.15 | — | — | — |
| | 4 | 12.56 | — | — | 87.47 | — | — | — | 442.82 | — | — | — |
| | 5 | 9.33 | — | — | 84.89 | — | — | — | 419.56 | — | — | — |
| IV | 1 | 22.74 | 0.019 | 0.68 | 64.70 | 0.002 | 5.81 | 0.17 | 506.84 | 0.0058 | 2.54 | 0.42 |
| | 2 | 19.53 | — | — | 63.09 | — | — | — | 485.45 | — | — | — |
| | 3 | 17.21 | — | — | 61.88 | — | — | — | 460.61 | — | — | — |
| | 4 | 14.63 | — | — | 61.27 | — | — | — | 458.08 | — | — | — |
| | 5 | 10.97 | — | — | 59.71 | — | — | — | 447.23 | — | — | — |

Note: stand type I is the pure larch forest; stand type II is the larch broadleaf mixed forest; stand type III is the Mongolian oak pure forest; stand type IV is the Mongolian oak broadleaf mixed forest. "—" means same parameters.

**Table 4.** Evaluation indices of the growth models by stand type.

| Stand Type | Model (12) | | Model (13) | | Model (14) | |
|---|---|---|---|---|---|---|
| | $R^2$ | RMSE | $R^2$ | RMSE | $R^2$ | RMSE |
| I | 0.9726 | 0.70 | 0.9794 | 1.03 | 0.8324 | 14.63 |
| II | 0.9725 | 0.70 | 0.9794 | 1.04 | 0.7482 | 34.24 |
| III | 0.9636 | 0.70 | 0.9723 | 1.40 | 0.8571 | 25.37 |
| IV | 0.9625 | 0.67 | 0.9787 | 1.27 | 0.9264 | 19.12 |

Note: $R^2$ is the coefficient of determination; RMSE is the root mean square error.

Using models (12), (13) and (14) and their parameter estimates, we plotted stand height, basal area and biomass as a function of stand age (*Year*) for different stand types at different site groups. Basal area and biomass are determined by stand age and stand density. For showing the relationship between basal area, biomass, and stand age more clearly in a 2-dimensional plan view, we assumed stand density (*S*) to be 1000. As we can be seen in Figure 2, models (12), (13) and (14) all exhibited a clear division by site groups, and the models are reasonable.

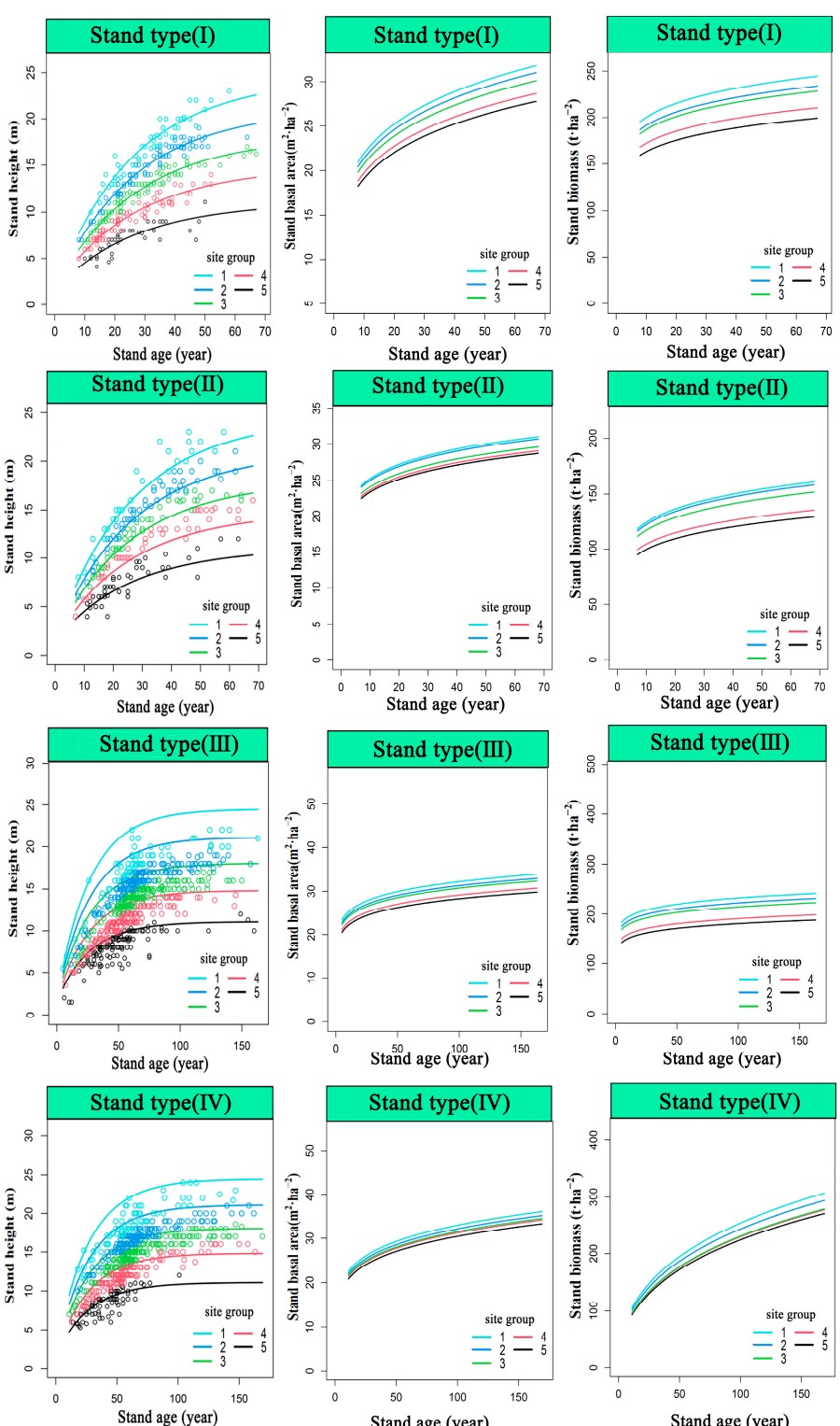

**Figure 2.** Growth curves for different site groups. For stand type, see definition in Table 2. For Site group, see definition in Section 2.3.2.

### 3.2. Computation of BPP by Stand Type

We plotted the BPP by stand type in the age range between 5 and 150 years (Figure 3), and there appeared to be a unimodal BPP curve for each site group and stand age in different age groups, which is inversely correlated. That is, the potential productivity decreased with stand age. The BPP of each stand type decreased with an increasing

site group, and BPP was the highest when the site group was 1 (site group 1 stands for the highest site quality and 5 stands for the lowest site quality). There was a negative relationship between the potential productivity and the stand class for a given age (i.e., the BPP decreased with an increasing site group).

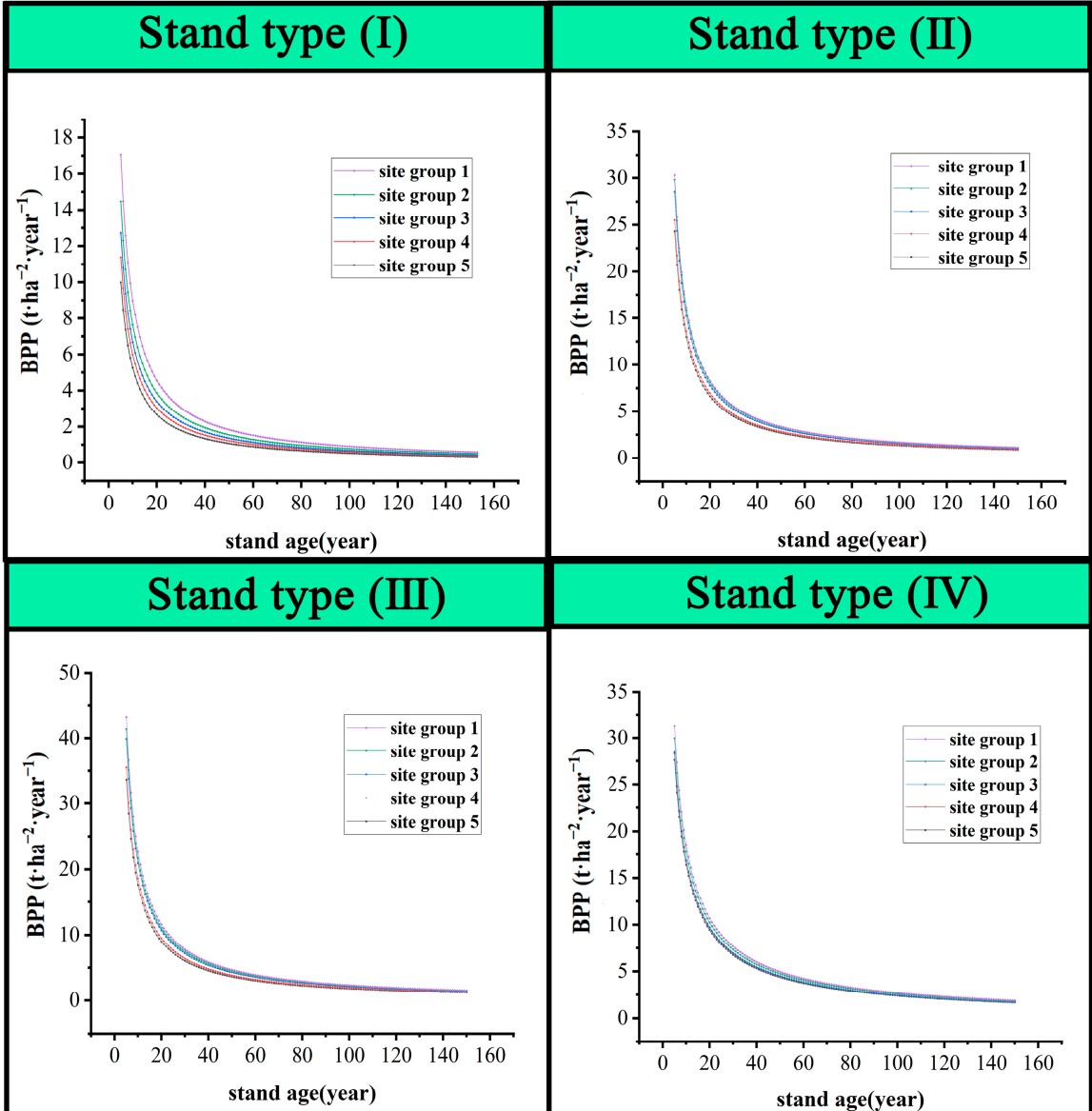

**Figure 3.** The biomass productivity potential (BPP) of different forest types on different site groups. For stand type, see definition in Table 2. For site group, see definition in Table 3.

### 3.3. BRP and BPI for Each Stand Type

The BRP reflects the realistic biomass accumulated in a stand during a year [27]. The realistic productivity of biomass was computed based on the stand age and stand density index. The computed biomass realistic productivity (BRP) and the available improved potential stand biomass (BPI) are shown in Figure 4. The trends of BRP and BPI seem to be similar for each stand type, with BRP showing an overall increasing trend with the stand age followed by decreasing trend, while the BPI decreases with stand age. The biomass potential productivity (BPP) is far from being brought into play at the middle-aged and young stage, which suggests that the earlier the forest quality is improved, the higher the stand productivity would be.

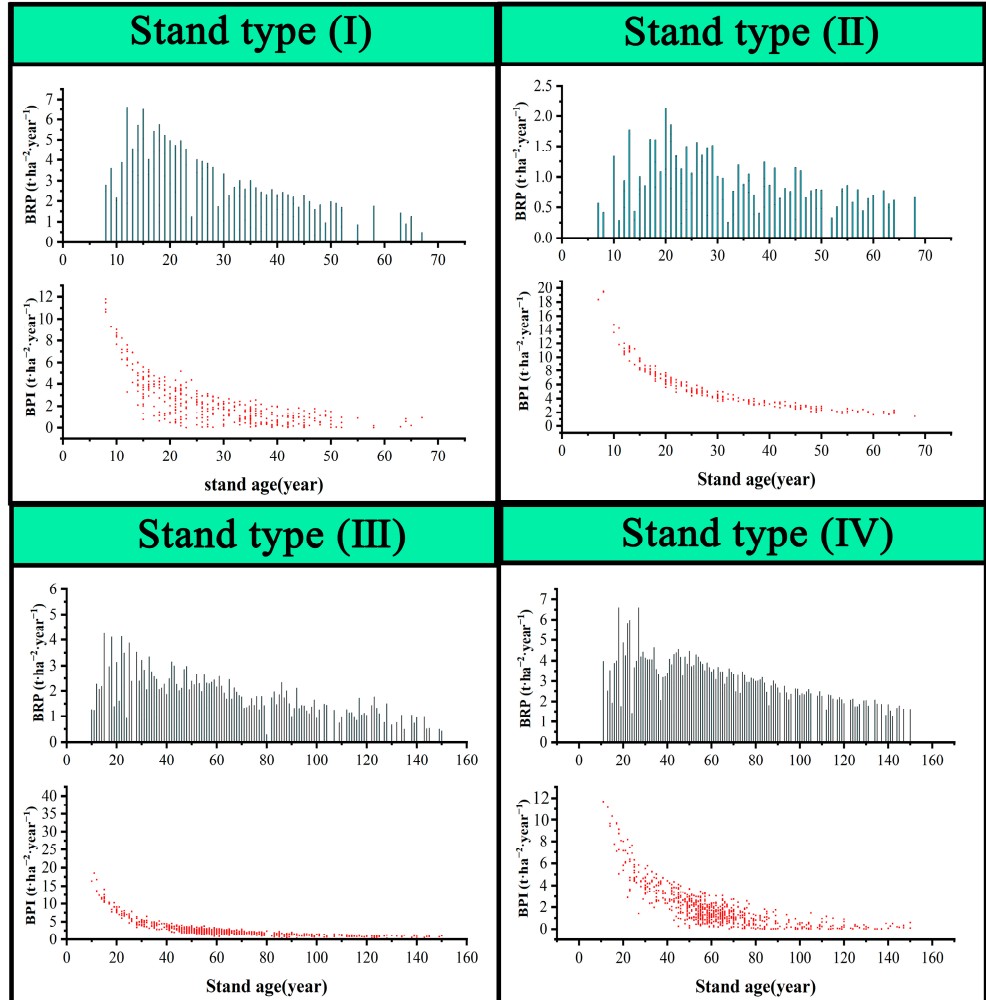

**Figure 4.** Biomass realistic productivity (BRP) and the available improved potential stand biomass (BPI) of each stand type. For stand type, see definition in Table 2.

### 3.4. Verification of BPP and Evaluation of Realistic Stands

The BPP and BRP for four forest types are shown in Figures 5–8. The difference between the BPP and BRP corresponding to each stand is large, the degree of difference is gradually decreasing as the stand age increases, and they are very close when the stand reaches the near-mature or mature stage. An identical pattern was observed for all site groups.

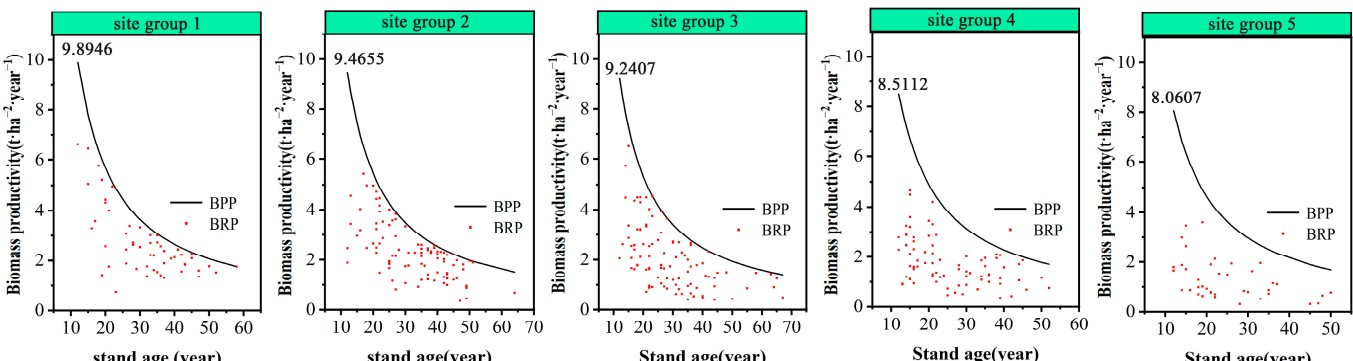

**Figure 5.** The biomass potential productivity (BPP) and biomass realistic productivity (BRP) of each sample plot against stand age (year) for group = 1, . . ., 5 for stand type I (pure larch forest). For site group, see definition in Table 3.

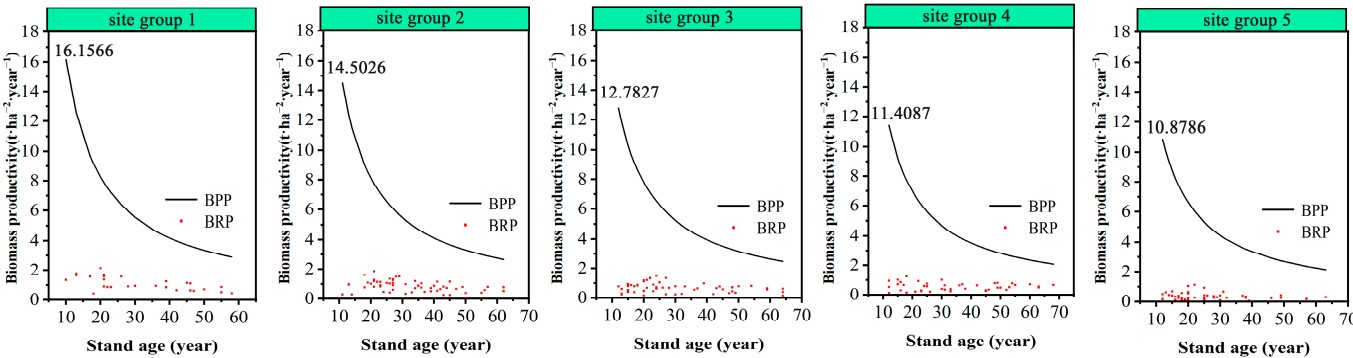

**Figure 6.** The biomass potential productivity (BPP) and biomass realistic productivity (BRP) of each sample plot against stand age (year) for group = 1, ..., 5 for stand type II (larch broadleaf mixed forest). For site group, see definition in Table 3.

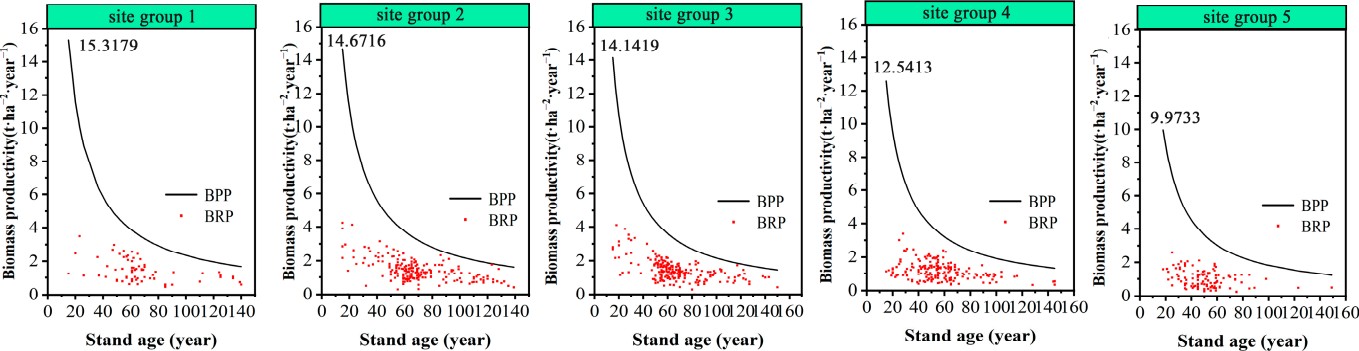

**Figure 7.** The biomass potential productivity (BPP) and biomass realistic productivity (BRP) of each sample plot against stand age (year) for group = 1, ..., 5 for stand type III (Mongolian oak pure forest). For site group, see definition in Table 3.

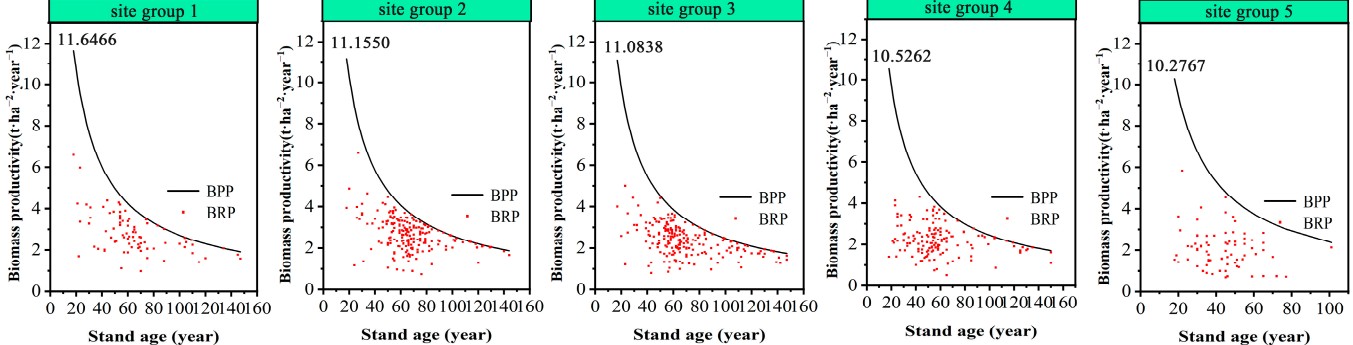

**Figure 8.** The biomass potential productivity (BPP) and biomass realistic productivity (BRP) of each sample plot against stand age (year) for group = 1, ..., 5 for stand type IV (Mongolian oak broadleaf mixed forest). For site group, see definition in Table 3.

### 3.5. Site Quality Evaluation at Reference Age

The potential productivity at a reference age can reflect the growth potential when the stand height tends to be stable and the mean growth is at its maximum (Table 5). In terms of the potential productivity of the biomass at reference age, the potential productivity levels of the four stand types were different at different site groups, and all showed that the potential productivity was the highest at site group 1 and the lowest at site group 5. Assuming a reference age of 30 for larch pure forest as an example, the potential productivity of the biomass corresponding to group 1 is 3.642 t·ha$^{-2}$·year$^{-1}$, which is 23% higher than that of site group 5 (2.966 t ha$^{-2}$ year$^{-1}$). Thus, site group 1 has a higher potential productivity and better stand quality.

**Table 5.** Biomass potential productivity by stand type for each site group.

| Stand Type | Potential Productivity | Site Group | | | | |
|:---:|:---|:---:|:---:|:---:|:---:|:---:|
| | | 1 | 2 | 3 | 4 | 5 |
| I | Biomass (t ha$^{-2}$ year$^{-1}$) | 3.642 | 3.483 | 3.400 | 3.132 | 2.966 |
| II | Biomass (t ha$^{-2}$ year$^{-1}$) | 5.552 | 5.459 | 5.230 | 4.668 | 4.452 |
| III | Biomass (t ha$^{-2}$ year$^{-1}$) | 7.763 | 7.436 | 7.167 | 6.371 | 6.037 |
| IV | Biomass (t ha$^{-2}$ year$^{-1}$) | 7.656 | 7.333 | 6.958 | 6.920 | 6.756 |

Note: For stand type, see definition in Table 2. For site group, see definition in Table 3.

## 4. Discussion

An accurate estimate of site productivity is an important part of formulating effective forest management strategies, and it is also the core basis for evaluating the basic status of sites in the ecological environment [32,33]. Biomass potential productivity and realistic productivity refer, respectively, to the maximum biomass accumulated in a stand and the realistic biomass accumulated in a stand during a year. We used the biomass potential productivity as a site quality evaluation index and ignored the traditional practice of selecting stands with good or poor site quality for site quality assessment. As mentioned in Section 1, we have combined the stand classification and site quality evaluation, introduced the detailed BPP computational processes, and have verified the rationality of this index by BRP, confirming a precise method that is applicable to forest site evaluation. The potential productivity of the biomass can be used as an indicator of site quality, providing reliable quantitative information for forest management decisions.

In the traditional methods of site quality evaluation, the dominant height at any reference age is chosen as the reference index. This method can be inaccurate when tree height growth is constrained by stand density or competition [34,35]. The potential productivity of the biomass as an evaluation index takes into account the influence of stand density and sets the maximum value of stand density, which is more reasonable for reflecting the actual situation of forest complexity in multi-aged and multi-layered mixed forests. In our study, the biomass potential productivity decreased with stand age in each forest type, and this decreasing pattern is consistent with the characteristics of the potential productivity (Figure 3). Because young stands would have a higher number of trees or a high stand density, consequently, there would be a higher potential productivity. That is, at a point in time, the potential productivity is a local extreme of the successive annual growth of the biomass. If the site conditions are favorable and the management practices employed are appropriate, the realistic productivity of the stand can approach or reach the potential productivity (Figure 4). The BPI is the available improved potential stand biomass. This indicator can also rationalize the priority of different stands for upgrading. For stands of the same age, the higher the BPI is, the higher the priority of promotion is; conversely, the lower the priority of promotion is (Figures 5–8). This is consistent with the trend of site quality in terms of the potential productivity evaluated in terms of the stand basal area approach [24]. This indicates that the management of young and middle-aged forests with good site quality should be emphasized in the process of forest nurturing. Our proposed concept of biomass potential productivity is reasonable, and our method for quantifying the site quality is feasible. In fact, the selection of evaluation indices is highly related to the purpose of forest site evaluation. The method of the basal area potential productivity emphasizes the stand diameter growth, which is more suitable for the evaluation of site quality aimed at producing stands with large-sized timber. However, for the special-purpose forests and public ecological welfare forests, where emphasis is placed on the forest biomass growth, the biomass productivity index can be intuitive and comprehensive to reflect the overall forest ecosystem quality. Therefore, it is necessary to choose the appropriate evaluation method according to the objectives.

The models we propose in the article are reliably applicable to other stand types in terms of species of interest and forested regions. The computation of the biomass potential

productivity requires, in addition to the given biomass growth model and its parameter estimates, a stand height growth model and a basal area growth model and their parameter estimates. All the growth models included in the biomass potential productivity model should have high accuracies in order to accurately estimate the forest site productivity. In our study, the whole-forest holistic model form was chosen, and both the stand basal area and stand biomass growth models were fitted with high accuracy for the actual stand conditions of the multi-aged and multi-layered mixed forests [36]. In the practical application, the stand basal area growth model or biomass growth model may be chosen based on the available stand type data. In conclusion, the biomass potential productivity approach is flexible enough to be applicable to a large-scale forestry, as data used to construct the biomass growth model are based on a large number of NFI plots, which cover the extensive forests (Figure 1). Our methods and models can provide both theoretical and practical knowledge, and their applicability will be useful for a larger scale site quality evaluation and even for tree species suitability evaluation.

We conducted this study under the assumption of equal annual growth of biomass regardless of the tree species and their growth behaviors. This is an ideal situation that is difficult to actually achieve. We were searching for the maximum value of the objective function in a given search interval for any stand density, which enables the computation of the potential productivity. The result obtained was that young stands would have a higher stand density, and thus have a higher potential productivity of the biomass. In contrast, near-mature and mature stands would have a lower stand density and lower biomass potential productivity. We may construct the segmentation function to solve this problem by assuming that a given forest type has the highest production potential at a given reference age. The maximum value of the objective function can be found for a given interval of the stand density, so that the potential productivity in the segmentation function can be obtained. In the actual forest growth, the natural process of self-thinning occurs when the age increases, and an increased age is accompanied by a decrease in the stand density of young and middle-aged stands. At the reference age, the stand density would be relatively stable [31,37]. Theoretically, the self-thinning model might improve the accuracy of the biomass potential productivity model if included; however, this requires the construction of the self-thinning model, which demands long-term time series data. The steps of computing biomass potential productivity may become more complicated when a self-thinning model is included. In our future studies, we are going to consider calculating the biomass potential productivity using the self-thinning model. In addition, as ecological problems and climate issues have attracted significant global attention, climate factors need to be included in the forest site quality evaluation methods [38,39].

## 5. Conclusions

In this study, we computed the potential productivity of the stand basal area and biomass for four stand types in the Jilin Province using four consecutively measured NFI data. We presented the growth model forms and their parameter estimates for stand height, basal area, and biomass, each of which had a reasonable fitting accuracy and model stability. The potential productivity is the local extreme value of the continuous annual growth of the forest biomass, and the realistic productivity of a stand can only be less than or equal to the potential productivity. If the site condition is good, the realistic productivity can approach or reach the potential productivity if the stand is in a good condition and the appropriate management measures are applied to the stands. In addition, the potential productivity of the middle-aged and young stands was significantly higher than that of the mature stands, and stands with better growth had a higher potential productivity than those with poorer stand growth. When evaluating the forest site quality, the biomass potential productivity at the reference age or close to the mature stand stage can often be used, which could provide a relatively high accuracy. Our potential biomass productivity concept is reasonable, and the method for its computation is feasible and applicable to multi-aged and multi-layered mixed forests. Forest managers should pay attention to the development of young and

middle-aged forests with better site quality for the improvement of forest production. Our study provides technical support for scientific forest management, which will have significant implications for the improvement of forest management and promotion of the modern forestry.

**Supplementary Materials:** The following supporting information can be downloaded at https://www.mdpi.com/article/10.3390/f15010023/s1, Supplementary Information S1: Method of biomass potential productivity (*BPP*); Supplementary Information S2: Method of biomass realistic productivity (*BRP*). All computations in this study were implemented in R4.3.0. The sample site factor computations such as biomass were implemented in the foreign package, the openxlsx package, the dplyr package, and the tidyr package. Stand height, basal area, and biomass growth models were used in the nlme package. The plotting was carried out with the ggplot package in R4.3.0 (www.r-project.org) and orgin 2022 software (www.origin.com/store/odc/store-plugin/views/checkoutLoading.html, accessed on 18 December 2023).

**Author Contributions:** X.Y., R.P.S., G.D., J.G. and L.F. (Linyan Feng). conceived the study. X.Y., L.F. (Liyong Fu) and L.P. performed the analysis and wrote the initial draft of the paper. All authors contributed to interpreting the results and the improvement of the paper. All authors have read and agreed to the published version of the manuscript.

**Funding:** This research was funded by the National Key Research and Development Program of China (Grant Number: 2022YFD2201005) and Accurate estimation of carbon sequestration potential of Plantation (Grant Number: DL2022052001L).

**Data Availability Statement:** The data used for the analysis and graphing in this paper are the 4-period national forest inventory data provided by the State Forestry and Grassland Administration (SFA) (https://www.forestdata.cn/index.html, accessed on 18 December 2023).

**Acknowledgments:** We also appreciate the valuable comments and constructive suggestions from three anonymous referees and the Associate Editor that helped improve the manuscript.

**Conflicts of Interest:** The authors declare no conflict of interest.

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
