# Peer review of "Evaluating Forest Site Quality Using the Biomass Potential Productivity Approach"

_forests, doi:10.3390/f15010023_

Round 1

Reviewer 1 Report

Comments and Suggestions for Authors

The Authors undertook and extensive study of the multiple forests based on the state inventory. The underlying assumptions of this study is to use average tree ages and sizes to estimate forest growth capabilities. The study look interesting and worth to be published in the journal.

The Discussion section mentioned that the presented model is a potential representation of the actual forests featuring different tree sizes and ages as a result different growth capabilities. It would be very beneficial for the manuscript if some estimation of the difference between the "simple" presented model and more sophisticated different-age model could be provided to grasp the order of the presented model uncertainty.

Sincerely,     

Author Response

Dear reviewers,

We express our sincere appreciations you editors and reviewers for constructive comments and suggestions to our manuscript (MS # forests-2689839). The comments and suggestions provided were considered helpful for use to improvement of the manuscript. We carefully revised the manuscript by incorporating all the suggestions provided by reviewers. We have also supplemented more details of the study methods to make our results more convincing to the reviewers.

In this revised version, all changes we made were highlighted with red-color. The point-by-point response to each of the reviewers’ comments are provided separately. Hopefully, our revised version will satisfy the reviewers.

Reviewer 2 Report

Comments and Suggestions for Authors

This is an innovative, interesting, and important investigation.

The reporting, however, is very far from satisfactory.

The most striking concern this reviewer finds relates to the shape of Figure 4. In ecology, productivity generally refers to the time rate of production. In Fig. 4, it appears to tend towards infinity with diminishing stand age. This can hardly be true, since at very young stands, there is not much green foliage.

If the shapes of the curves are correct, the concept “BPP” must refer to something else than “biomass potential productivity”. The units in the graphic (and in Table 5) indicate it possibly does refer to something else. However, this reviewer does not get it clear from Figure 3 what it refers to. This is partially because it is hard to read the Figure without a magnifier glass. (The same applies to most of the Figures.) There is a definition on lines 60 to 61. However, the curve shapes, and the units, do not appear to match with this definition.

Further remarks:

The Authors probably should include a more detailed description of the techniques listed in lines 42…45  of the Introduction. Based on the current text, it is not possible to understand the text written by the Authors, without studying the given references.

The symbols used in Table 1 do not seem to converge.

This reviewer does not understand the first line of Eq. (1).

Line 143: what “site groups”?

It appears that Equations (5) to (9) correspond to height, basal area, …, biomass models – not to corresponding growth models. Growth models should refer to time derivatives. Or are these models of accumulated growth?

Author Response

Dear reviewer,

We express our sincere appreciations you editors and reviewers for constructive comments and suggestions to our manuscript (MS # forests-2689839). The comments and suggestions provided were considered helpful for use to improvement of the manuscript. We carefully revised the manuscript by incorporating all the suggestions provided by reviewers. We have also supplemented more details of the study methods to make our results more convincing to the reviewers.

In this revised version, all changes we made were highlighted with red-color. The point-by-point response to each of the reviewers’ comments are provided separately. Hopefully, our revised version will satisfy the reviewers.

Round 2

Reviewer 2 Report

Comments and Suggestions for Authors

The Authors have provided reasonable additional comments, as well as editions.

It appears, however, that there is a significant problem related to Fig. 3 and Eq. 10.

In the absence of the site density index in Eq. (10), Eq. 3 should show a non-monotonic evolution of biomass production rate as a function of stand age, which would make sense.

Introduction of the site density index appears to confuse things. Its application apparently produces incorrect results.

It is hard to see any other solution but rejection of the paper.

Author Response

Dear reviewer,

We feel great thanks for your professional review work on our manuscript. Our manuscript (MS # forests-2689839), which was rejected. Because of some issues due to the absence of a detailed description of the methods we employed in our study. We have added a detailed methodological description on such issues in the current manuscript together with detailed mathematical procedures for obtaining the results (as solutions to the problematic issues). As originally or manuscript was designed to submit to Forests, we still want to publish this in this journal. We would like to request you to consider for further editorial processing of our resubmission. Our modified manuscript texts are shown in red color. We ensured the accuracy of all the equations and information provided in a new submission. We hope that the revised manuscript will be adequate for publication in the Forests. Thanks!

Sincerely yours,

Xingrong Yan
